# The Role of Selected Interleukins in the Development and Progression of Multiple Sclerosis—A Systematic Review

**DOI:** 10.3390/ijms25052589

**Published:** 2024-02-23

**Authors:** Cezary Grunwald, Anna Krętowska-Grunwald, Edyta Adamska-Patruno, Jan Kochanowicz, Alina Kułakowska, Monika Chorąży

**Affiliations:** 1Department of Neurology, Medical University of Bialystok, Marii Skłodowskiej-Curie 24A, 15-276 Białystok, Poland; jan.kochanowicz@uskwb.pl (J.K.); alina.kulakowska@umb.edu.pl (A.K.); 2Department of Pediatric Oncology and Hematology, Medical University of Bialystok, Jerzego Waszyngtona 17, 15-274 Białystok, Poland; anna.kretowska-grunwald@umb.edu.pl; 3Clinical Research Center, Medical University of Bialystok, Marii Skłodowskiej-Curie 24A, 15-276 Białystok, Poland; edyta.adamska-patruno@umb.edu.pl

**Keywords:** interleukins, cytokines, multiple sclerosis, autoimmune disease, experimental autoimmune encephalomyelitis

## Abstract

Multiple sclerosis is a disabling inflammatory disorder of the central nervous system characterized by demyelination and neurodegeneration. Given that multiple sclerosis remains an incurable disease, the management of MS predominantly focuses on reducing relapses and decelerating the progression of both physical and cognitive decline. The continuous autoimmune process modulated by cytokines seems to be a vital contributing factor to the development and relapse of multiple sclerosis. This review sought to summarize the role of selected interleukins in the pathogenesis and advancement of MS. Patients with MS in the active disease phase seem to exhibit an increased serum level of IL-2, IL-4, IL-6, IL-13, IL-17, IL-21, IL-22 and IL-33 compared to healthy controls and patients in remission, while IL-10 appears to have a beneficial impact in preventing the progression of the disease. Despite being usually associated with proinflammatory activity, several studies have additionally recognized a neuroprotective role of IL-13, IL-22 and IL-33. Moreover, selected gene polymorphisms of IL-2R, IL-4, IL-6, IL-13 and IL-22 were identified as a possible risk factor related to MS development. Treatment strategies of multiple sclerosis that either target or utilize these cytokines seem rather promising, but more comprehensive research is necessary to gain a clearer understanding of how these cytokines precisely affect MS development and progression.

## 1. Introduction

Multiple sclerosis (MS) is a disabling progressive autoimmune disease of the central nervous system which is characterized by axonal demyelination [1,2,3]. Nearly 3 million people are currently living with MS worldwide, and, as with many inflammatory disorders, the majority are women [4,5]. The clinical manifestation of MS was traditionally divided into four subtypes, with around 85% patients suffering from relapsing–remitting multiple sclerosis (RRMS). The remaining 15% present either with primary or secondary progressive (SPMS) or progressive relapsing multiple sclerosis. In time, most RRMS patients advance to the slow-progressing SPMS type; however, novel disease-modifying treatment has significantly prolonged this progression [6,7]. This classification, however, did not take into consideration disease complexity and the overlapping characteristics of selected phenotypes; therefore, a 2013 consensus defined clinically isolated syndrome and radiologically isolated syndrome as potential disease courses [8,9].

The diagnosis of multiple sclerosis is based on thorough symptom analysis, neurological examination and additional investigations, which include magnetic resonance imaging performed accordingly to the revised McDonald criteria (2017) and cerebrospinal fluid examination of the presence of oligoclonal bands [10,11]. The clinical presentation of MS varies depending on the location of demyelinating lesions in the central nervous system (CNS) and may manifest as vision impairment, ataxia, muscle deficiency and more. Frequently, the onset of the disease presents as an isolated syndrome—optic neuritis (e.g., unilateral short-term blindness), isolated spinal cord syndrome (e.g., self-limiting sensory myelitis) or isolated brain stem syndrome (e.g., abducens nerve palsy) [12]. In RRMS, moments of symptom aggravation intertwine with stable disability [2,10]. 

With multiple sclerosis being to date an incurable disease, the treatment of MS relies mainly on limiting relapses and slowing down the process of motor and intellectual impairment [13]. Interestingly, the predisposition to acquiring MS may to an extent be hereditary, with immediate family members being at a much higher risk of developing the disease. Several studies have additionally identified gene polymorphisms of selected cytokines, including IL-2, IL4, IL-6, IL-7 and IL-13, as pathognomonic for MS [14,15,16,17,18]. Although immunomodulation, i.e., interferon (IFN-b) and glatiramer acetate, is the standard first-line therapy for MS, the use of more novel disease-modifying drugs such as natalizumab and fingolimod as the second-line treatment has become standard practice [19,20,21]. A better understanding of the B cell’s role in MS pathophysiology led to the shift from sole T-cell-focused therapies to the introduction of anti-CD20 monoclonal antibodies ocrelizumab and ofatumumab into MS treatment [22]. 

The ongoing acute inflammatory reaction developing owing to the disfunction of the blood–brain barrier (BBB) is thought to be the underlying cause of MS [6,7]. This disruption can be visualized with magnetic resonance imaging with the use of gadolin early in the disease pathogenesis [23]. The breakage in the BBB is, among others, a result of endothelial cell disruption by the increased level of leukocyte adhesion molecules enabling an amplification of leukocyte transmigration [24]. These consist mainly of CD4+ Th1 and Th17 lymphocytes, which are responsible for releasing myelinotoxic cytokines such as IL-33, leading to demyelination [25,26]. Upon interaction with monocytes, which are the dominant cell population in CNS infiltration, the tight junctions of the BBB endothelial cells become increasingly permeable due to reactive oxygen species excretion. This dysregulation of the BBB was found to be additionally caused by IL-17. The influx of immune cells into the CNS therefore results in the release of various immune mediators, including proinflammatory interleukins, intensifying the immune response in the CNS, causing demyelination and neurodegeneration [24,27]. Interestingly, the onset of MS does not seem to be strictly associated with peripheral blood immune cell infiltration, but is additionally influenced by the discrepancies inside the “MS brain”, especially microglia/macrophages [27]. 

Cytokines have been assigned an immunoregulatory role, and, given the inflammatory basis of multiple sclerosis, numerous studies have investigated their impact on the disease pathogenesis and progression [7,17,28]. Interestingly, the eminent disharmony between Th1 and Th2 cytokines may be identified as the backbone of MS pathogenesis, with more current studies focusing additionally on the analysis of the sub-populations of T- helper cells [29,30]. The ongoing cytokine-mediated autoimmunological process seems to be the underlying cause of multiple sclerosis; thus, maintaining the balance between pro- and anti-inflammatory activity seems crucial in preventing disease progression [31]. Taking this into consideration, a thorough investigation into the exact role of cytokines in MS is crucial in order to determine their potential role in the development of novel therapeutic targets. 

In this article, we review recent data regarding the role of selected interleukins in the regulation of the immune system in multiple sclerosis development and progression, which can be a vital source of information in light of ongoing investigations into novel treatment strategies of multiple sclerosis that either target or utilize cytokines. Understanding the exact impact of interleukins on MS development is essential prior to their implementation in therapy given their not-always-ambiguous role, clearly exhibited in this review. 

## 2. Materials and Methods

An extensive search was last carried out in February 2024 in the PubMed online electronic database using the search terms “multiple sclerosis” and (“interleukins” or “IL-2” or “IL-4” or “IL-6” or “IL-10” or “IL-13” or “IL-17” or “IL-21” or “IL-22” or “IL-33”). There were no specified time limits imposed on the selected publications. After initial selection, a total of 172 articles were retrieved, and, following the PRISMA guidelines, selected articles were included in this study. Ten duplicate articles were identified and subsequently removed. Eighteen articles were removed from consideration because they contained irrelevant data for the purpose of this review, two were published in languages other than English and nineteen were not accessible. In the end, a total of 135 articles were analyzed for this review. Figure 1 illustrates the process of selecting scientific papers for this article.

## 3. Results and Discussion 

### 3.1. Interleukin 2 (IL-2)

IL-2 is one of the key factors in maintaining immunological tolerance. Produced predominantly by Th1 lymphocytes, they act as immune modulators by promoting the differentiation of immature T cells into regulatory T cells and inhibiting Th17 cell development, which results in prevention of autoimmune diseases [33,34]. A study by Sadlack et al. involving IL-2-deficient mice demonstrated their tendency to develop autoimmunity in the form of bowel inflammation similar to ulcerative colitis [35]. Contrary to murine species, deletion of IL-2R in humans results not only in loss of immune tolerance but also in immunological deficiency [36]. Petrzalka et al. associate increased IL-2 CSF levels with a worse clinical outcome with a shorter initial remission interval [37]. IL-2 might exert a cytotoxic effect and promote inflammation on MS lesions through IFN-γ released by activated natural killer cells [38].

Several studies have identified increased IL-2 levels in the peripheral blood and the cerebrospinal fluid (CSF) in MS patients compared to healthy controls [34,39]. Sharief et al. correlated elevated intrathecal IL-2 and soluble IL-2R (sIL-2R) with the level of blood–brain barrier disfunction, suggesting their possible role in BBB impairment [40]. In another study by Sharief et al., they analyzed the relationship between this cytokine and its circulating receptor and MS disease severity and found higher CSF IL-2 and sIL-2R levels in patients during an MS relapse compared to those in remission [41]. Interestingly, a study by Gilio et al. suggests that decreased CSF IL-2 due to preventative physical activity may lead to an improved emotional state in both human and murine subjects, reducing anxiety and depression-like behavior [42]. Contrary to previously stated studies, Carbone et al. found depleted levels of serum IL-2 in patients with RRMS and assessed the proliferation of Tregs in RRMS to be partially independent of IL-2 secretion [43].

Elevated levels of soluble IL-2Rα chains (sIL-2RA) can be found in the peripheral blood of patients with autoimmunological diseases. Although not fully understood, sIL-2RA seems to have an antagonistic effect on IL-2, inhibiting IL-2-mediated T cell responses in in vitro models [44]. Russel et al. exhibited the association between aggravated experimental autoimmune encephalomyelitis (EAE) and increased serum sIL-2RA with concomitant enhanced Th1 and Th17 response and CNS infiltration in murine species [45]. Lu et al. correlate higher sIL-2RA levels with increased susceptibility to multiple sclerosis [46]. 

A meta-analysis by Wang et al. regarding genetic susceptibility to MS revealed interleukin 2 receptor alpha (IL2RA) gene polymorphism *rs2104286* to be related to greater risk of developing MS in both the Asian and Caucasian patient populations. This association was additionally observed in regards to IL-2RA gene polymorphism *rs12722489*, but not in the Asian population [17]. On the contrary, two independent studies by Ali Shokrgozar et al. and Alsahebfosoul et al. did not find associations between IL-2/IL-2RA gene polymorphisms and predisposition to MS in the Iranian population [38,47]. Buhelt et al. did not observe the correlation between the IL2RA SNP *rs2104286* and biomarkers of inflammation [48].

### 3.2. Interleukin 4 (IL-4)

IL-4 is a cytokine that plays a key role in type 2 immune response, especially in the context of allergic inflammation or parasitic invasion [49,50]. It serves as an immunoregulatory agent, associated with mediating allergy reactions through encouraging B cell production of Immunoglobulin E (IgE) and Th2 lymphocyte development [51,52]. IL-4 has been investigated for its involvement in the pathogenesis of numerous diseases including inflammatory arthritis, allergic asthma and several cardiovascular diseases [53,54,55]. Despite the existence of the blood–brain barrier, type 2 immunity seems pivotal in maintaining central nervous system homeostasis. Interestingly, T cells were found to play a role in cognitive skill development, with meningeal IL-4 depletion causing decreased learning ability in a murine model. This opens the window for further research into novel immunotherapies for cognitive impairment in the state of immunodeficiency [56,57]. 

Given the potential impact of type 2 immunity on the CNS immunological stability, numerous studies have moreover investigated the role of IL-4 in the pathogenesis of multiple sclerosis. According to Ponomarev et al., cerebral IL-4 seems to play a role in MS regulation as IL-4 deficiency in the CNS resulted in EAE exacerbation despite maintaining a normal IL-4 peripheral level [58]. On the other hand, Tahani et al. reported the serum level of IL-4 in multiple sclerosis patients to be nearly three times higher than that of the healthy control group with no significant difference between males and females [7]. The significance of IL-4 in the development of MS was suggested in several studies, with the polymorphism of IL4 *rs2243250* being associated with MS incidence in both dominant and allele models [7,15,59]. Interestingly, the postmortem analysis of MS patients’ brain specimens revealed an increased level of IL-4 immunoreactivity in astrocytes located in areas of chronic lesions with decreased cellularity. The elevation, however, could also be observed in the location of cerebral infarction-induced proliferation of glial cells, indicating lack of specificity for multiple sclerosis [59].

### 3.3. Interleukin 6 (IL-6)

IL-6 is a proinflammatory cytokine, crucial in the development and modulation of numerous immune cells, that is often increased in autoimmunological diseases [37,60,61]. IL-6 stimulates the production of demyelinating agents contributing to the pathogenesis and progression of multiple sclerosis [62]. According to Barr et al., IL-6 B cell secretion was significantly higher in MS patients compared to the healthy control group. Notably, after implementation of therapy resulting in B cell depletion, the concomitant decrease in disease severity and the level of IL-6 was observed. This may be attributed to IL-6-mediated stimulation of pathogenic T cells, mainly Th17 lymphocytes, which are pivotal in MS development [63,64]. Interestingly, according to Patanella et al., higher levels of IL-6 were observed in the peripheral blood of relapsing–remitting MS patients with poor cognitive performance based on their Mini Mental State Examination score (MMSE) [65].

Vandebergh et al. found patients with increased IL-6 signaling to have a higher risk of developing MS and additionally exhibited some associations of MS with a genetic predisposition to elevated body mass index (BMI) [66]. The analysis of the SNP *rs1818879* of the IL-6 gene revealed its association with both the incidence of active lesions in the brain of MS patients and the increase in cytokines mediating the neuroinflammatory process in MS pathogenesis [18].

Several studies have investigated the possible use of anti-IL-6R (tocilizumab) in the treatment of multiple sclerosis. Staley et al. have proven tocilizumab to be successful in limiting EAE progression in a murine model [67]. Zheng et al. correlated higher soluble IL-6R level with a decreased risk of MS, confirming the basis for the implementation of tocilizumab as a novel MS therapy [68]. Reports on actual clinical use of tocilizumab in MS are quite scarce. Sato et al. described a case of a 53-year-old Japanese woman with a concomitant incidence of multiple sclerosis and rheumatoid arthritis who, after treatment with tocilizumab, has been in remission for over 5 years (as of 2014) [69]. Although tocilizumab therapy for MS seems promising, Janssens et al. suggest caution in its implementation due to several reports on the neuroprotective influence of IL-6 [64]. 

### 3.4. Interleukin 10 (IL-10) 

IL-10, as a pivotal anti-inflammatory cytokine, plays a key role in immunosuppression, acting mainly through antigen-presenting cell inhibition. Upon inflammatory conditions, it is predominately produced by stimulated leukocytes [60,70,71]. It is a crucial factor in the regulation of autoimmunity, as IL-10- or IL-10R-deficient mice seem to acquire spontaneous autoimmune diseases, e.g., colitis [72]. IL-10 has also been under investigation as a possible prognostic factor in oncological malignancies, as the presence of this cytokine in the tumor microenvironment seems to be correlated with inferior prognosis [73,74]. Despite this, a study by Mumm et al. revealed that IL-10 seems to promote CD8+ T cells and enhance their cytotoxic activity, resulting in decreased carcinogenesis [75]. IL-10 seems to be an ideal target for immunotherapy in numerous diseases such as inflammatory bowel disease, psoriasis, rheumatoid arthritis or various solid tumors. However, despite a broad number of clinical trials focusing on the subject, no approved treatment is yet available [72,76].

In multiple sclerosis, IL-10 seems to exert a protective effect against disease progression, with PBMC derived from MS patients secreting a decreased amount of IL-10 compared to a healthy control in an in vitro model [60]. In a murine model by Bettelli et al., IL-10-deficient mice were more likely to exhibit a severe form of autoimmune encephalomyelitis compared to both mice lacking IL-4 and a control wild-type group [77]. Similarly, prior to this, Rott et al. assigned IL-10 a preventative role in EAE development in Lewis rat species [78]. In a more recent study, Pennatti et al. attribute this phenomenon to IL-10+ regulatory B-cell-associated restoration of the CNS immune environment and consequent remyelination by oligodendrocyte precursor cells [79]. However, on a molecular level, no significant correlation between IL-10 gene polymorphisms and MS occurrence was found by Ramakrishnan et al. [80]. 

Interestingly, patients with a history of an Epstein–Barr infection seem to be more prone to the development of multiple sclerosis in an IL-10-dependent mechanism. Through a high set point of latent EBV-infected B cell load in the CNS, cellular IL-10 and its EBV-encoded homolog create pathogenetic T cell responses, resulting in an inflammatory imbalance, which may act as a factor in MS activation [81].

Although direct administration of IL-10 seems to be a promising strategy in the treatment of several systemic diseases due to its limiting characteristics, i.e., short half-time, IL-10-secreting B cell manipulation seems to be a better approach in MS [82]. It is noteworthy that therapy with glatiramer acetate, a currently widely approved treatment for multiple sclerosis, was found to be related to IL-10 elevation [83]. 

### 3.5. Interleukin 13 (IL-13)

IL-13 as an immunomodulatory, Th2-derived cytokine has to date been commonly assigned an anti-inflammatory role, especially in the autoimmune and allergic settings. It seems to play a role in the pathogenesis of such medical conditions as inflammatory arthritis, atopic dermatitis, colitis, COVID-19 infection and multiple sclerosis [14,53,84,85,86]. However, the exact role, either positive or negative, of IL-13 in the development and progression of multiple sclerosis still remains ambiguous. The incidence of the IL-13 receptor on the dopaminergic neurons might indicate its role in the dopaminergic pathway [87]. A genetic association with susceptibility to or prevention of multiple sclerosis was evaluated by Seyfizadeh et al., who analyzed single nucleotide polymorphisms of IL-13 and found significant connection between certain IL-13 SNPs (*−1112CT*, *−1512AC* and *+2044GA*) and MS predisposition and the age of onset [14]. Interestingly, a study analyzing the *R130Q* IL-13 SNP did not make a note of such correlation [88]. 

According to Ghezzi et al., patients during an MS relapse exhibited significantly higher levels of IL-13-producing CD4+ lymphocytes in the cerebrospinal fluid than those in remission. Interestingly, there seemed to be a positive correlation between IL-13-producing T lymphocytes and the patient disability measured according to the Expanded Disability Status Scale (EDSS). The authors also seem to have observed an IL-13 elevation in the postmortem meninges of active lesions in MS patients [89]. On the other hand, Rossi et al. observed an increase in the CSF IL-13 in relapsing patients as well as in those in remission. The authors provide evidence for the neuroprotective role of IL-13, with this cytokine being strongly correlated with amyloid-b1-42 level in the CSF and neuronal cohesion, identified by optical coherence tomography (OCT). Despite this, Rossi et al. did not achieve exhibiting a correlation between IL-13 CSF levels and MS progression based on the EDSS; the correlation between the Multiple Sclerosis Functional Composite (MSFC) scale was positive but not significant [90]. Interestingly, a study by Cash et al. confirms that human recombinant IL-13 might exert a protective effect and suppress experimental autoimmune encephalomyelitis in a murine model [91]. These discrepancies indicate a great need for further research into the exact role of IL-13 in pathogenesis of multiple sclerosis. 

### 3.6. Interleukin 17 (IL-17)

IL-17, a proinflammatory cytokine mainly produced by immune cells, i.e., Th17 lymphocytes, plays a valid role in controlling inflammatory responses in the local tissue environment [92]. IL-17A and IL-17F are IL-17 isoforms most commonly analyzed in the context of pathogenesis of autoimmune disorders. Although they often may be treated as one, IL-17, this might be an overgeneralization as they seem to exert a similar but not identical proinflammatory effect [93].

According to Schofield et al., both serum and CSF levels of IL-17AA were higher in patients diagnosed with RRMS than in healthy subsets. The authors suggest, however, that, due to IL-17 concentrations detected at ug–pg/mL levels, highly sensitive methods should be used for the assessment [94]. Interestingly, however, Hartung et al., upon analyzing more than 230 patients with RRMS treated with interferon beta-1b, did not observe any significant changes in IL-17F serum levels between the time of diagnosis and early treatment, putting into question the role of IL-17F as a biomarker of therapy response [95]. In addition to Th17 cell subsets, astrocytes located in active MS lesions were found to have the ability to produce IL-17 under adequate conditions [96]. Similar findings were made by Tzartos et al., who observed an increase in IL-17-positive CD4+ and CD8+ T lymphocytes in acute MS areas [97]. Interestingly, according to a study by Acosta-Rodriguez et al., only CD4+ lymphocytes (Th17), and not CD8+ ones, were able to secrete IL-17 in the blood of healthy human subsets, drawing the hypothesis that they may be only associated with the microenvironment of the CNS [98]. Recent studies, however, associate the presence of IL-17+CD8+ T lymphocytes (Tc17) with the sites of active inflammation [99]. The increased expression of IL-6 in astrocytes in the presence of IL-17 results in elevated chemokine ligand 20 (CCL20) production, which encourages T cell migration into the central nervous system, emphasizing the possible role of IL-17 in MS pathogenesis [100]. The role of IL-17 in the development of multiple sclerosis was also confirmed by Kang et al., who determined that deficiency of Act-1, an IL-17 receptor complex adaptor crucial for IL-17 signaling, resulted in weakened inflammatory response, causing impaired development of experimental autoimmune encephalomyelitis, in an in vitro murine model for MS [92,101]. 

### 3.7. Interleukin 21 (IL-21)

IL-21 is known for its multi-functional role as an immune-cell-regulating cytokine, essential in the maintenance of immunological homeostasis, acting through both pro- and anti-inflammatory responses. It was found to drive the growth of progenitor IL-10, producing B cells into maturity through CD40 assistance [102,103,104]. The presence of IL-21 seems to lead to the enzymatic activation of regulatory B cells (Bregs), making them capable of T cell proliferation limitation through granzyme-dependent T cell receptor inactivation [105]. Additionally, Nurieva et al. exhibited a high expression of IL-21 in Th-17 lymphocytes in a murine model [106]. According to several studies, the IL-21 level was found to be increased in a number of autoimmunological diseases, including systemic lupus erythematosus, rheumatoid arthritis and multiple sclerosis [107,108,109]. 

Christensen et al. found elevated IL-21 and IL-21R on CD4+ lymphocytes in the peripheral blood of patients with progressive MS [110]. It is noteworthy that Ali Abdulla et al. suggested a potential role of the IL-21 wild homozygous genotype (*rs2055979*) in the pathogenesis of MS [111]. Furthermore, Gharibi et al. found a significant correlation between IL-21 mRNA and serum levels and the severity of multiple sclerosis assessed according to the EDSS scores, pointing to the proinflammatory role of IL-21 [109]. Interestingly, the inhibition of IL-21 resulted in mice protection against EAE by way of Th17 depletion, making this cytokine an ideal target for immunotherapy against multiple sclerosis [106]. Edo et al. investigated the possible therapeutic blockage of the IL-21–IL-21R axis with specific consideration of time-dependent treatment initiation, which resulted in promising results in the context of MS prevention [112]. On the other hand, injection of IL-21 prior to EAE induction in a murine model resulted in an increase in disease severity compared to the control group. When the cytokine was administered after EAE development initiation, there seemed to be no significant difference in the EAE progression between the studied groups, regardless of the lack of disease symptoms [113]. This suggests that, although IL-21 seems to play a crucial role in the pathogenesis of EAE, extensive research regarding the moment of treatment initiation should be undertaken before considering targeted therapy against IL-21. 

Alemtuzumab is a monoclonal anti-CD52 antibody which, in recent years, has been under investigation as a possible therapeutic agent for multiple sclerosis and has shown promising results in prolonging relapse- and progression-free survival [114]. Interestingly, an increased IL-21 level can additionally act as a predictive factor of secondary autoimmune disorder incidence in MS patients after alemtuzumab treatment [104].

### 3.8. Interleukin 22 (IL-22)

The role of IL-22, a proinflammatory member of the IL-10 cytokine family, in the pathogenesis of multiple sclerosis remains to date ambiguous [115]. It can act as a pathological agent in several autoimmunological diseases and exert a tumor-promoting effect in numerous cancer types, i.e., breast, colon and lung cancers [116]. It is highly expressed by Th17 cells; however, IL-22 can additionally be secreted by Th22, a fairly newly discovered T lymphocyte CD4+ subset which resides in tissues [117]. 

According to several studies, IL-22 seems to play a causative role in the development of MS, given the increase in IL-22 peripheral blood and cerebrospinal fluid level in relapse patients compared to both those in remission and healthy subsets [115,118,119,120]. Almolda et al. aimed to evaluate the dynamic changes of cytokine levels at subsequent phases of EAE in a murine model and found an increase in IL-22 during active phases of the disease and a depletion in remission [121]. Similarly to Th17, Th22 might disturb the blood–brain barrier, leading to inflammation through recruitment of CD4+ T lymphocytes, which could be potentially meaningful in the pathogenesis of multiple sclerosis [118,122].

Interestingly, a study on a Scandinavian population identified a single nucleotide polymorphism (SNP) found on the *IL-22RA2* gene, which encodes IL-22 binding protein (IL-22BR), as a possible risk factor related to MS development [123]. These results were confirmed by Lindahl et al. on a rat and murine models [124]. Despite the suggested causative role of IL-22 in MS development, Kreymborg et al. found no association between the knockout of IL-22 in mice and the disease onset [125]. The deletion of IL-22BR (an IL-22 antagonist), however, resulted in milder EAE presentation, which Lindahl et al. explain as a possible uninhibited IL-22 pathway. This might translate into a possible protective impact of IL-22 on neuroinflammatory processes. [124] Moreover, the in vitro addition of IL-22 seems to improve the survival of astrocyte cultures in both healthy and MS subjects [119]. 

### 3.9. Interleukin 33 (IL-33)

IL-33, a relatively newly discovered (i.e., 2005) cytokine belonging to the IL-1 cytokine family, has been under investigation as a possible novel target for therapy of autoimmune diseases with its ligand interleukin 1 receptor-like 1 (ST2), a marker associated with Th2 lymphocytes [26,126]. Although usually associated with proinflammatory activity, several studies have additionally named its protective role in various diseases of the cardiovascular system [126,127]. 

With high expression of astrocyte-activating IL-33 in the central nervous system, this cytokine seems to play an important role in the demyelination process through the activation of proinflammatory cytokines such as IL-6, tumor necrosis factor (TNF) and transcription factors, i.e., nuclear factor kappa B (NF-κB)—the baseline of the pathogenesis of multiple sclerosis [26,128,129]. IL-33 therefore acts as an alarming factor which, upon dysregulation of cell stress and death pathways, activates the immune system, assigning astrocytes a potential role of “immune regulators of the CNS” [130,131]. 

According to Alsahebfosoul et al., IL-33 plasma levels were found to be higher in relapsing–remitting multiple sclerosis patients compared to a healthy control. Interestingly, those patients who were already being treated with interferon beta-1a (IFN-β) presented with lower levels of IL-33 than those who were untreated, which emphasizes the pivotal role of IL-33 in the ongoing inflammatory process [126,132]. Similar results were obtained by Christophi et al., who additionally evaluated an increased level of IL-33 in the white matter and plaque areas from MS brain samples [133]. The unambiguous difference in plasma IL-33 levels between patients with MS and healthy controls, as well as its decrease in patients currently under treatment, manifests a possible beneficial role of IL-33 in monitoring disease activity [131,134]. Interestingly, according to Natarajan et al.’s in vitro studies, IL-33 might, on the other hand, promote myelin repair, exhibiting its neuroprotective role [135]. Therefore, the exact role—either beneficial or causative—of the IL-33 in the pathogenesis of MS still remains ambiguous.

## 4. Conclusions

Multiple sclerosis is a progressive autoimmune disease with currently no available treatment resulting in a successful cure. It affects mainly young adults, leaving them burdened with numerous symptoms and disability. Currently approved therapies for MS rely mainly on non-specific anti-inflammatory treatment. The continuous autoimmune response driven by cytokines appears to be the fundamental factor behind the development of multiple sclerosis; therefore, it is vital to preserve a delicate but still to date not fully understood balance between inflammatory and anti-inflammatory mechanisms to halt the advancement of the disease (Table 1). As mentioned in this review, selected interleukins have been shown to exhibit an only promoting (IL-4, IL-17, IL-21) or exclusively protective (IL-10) role in MS development, with some cytokines being identified as exhibiting both (IL-2, IL-6, IL-13, IL-22, IL-33). Given the impact of these selected immune molecules in the course of multiple sclerosis, studies into novel therapies either targeting or utilizing these cytokines seem promising. However, further in-depth research is required to better understand the exact impact of these cytokines in MS development as numerous cytokines were found to exhibit an ambiguous role. 

## Figures and Tables

**Figure 1 ijms-25-02589-f001:**
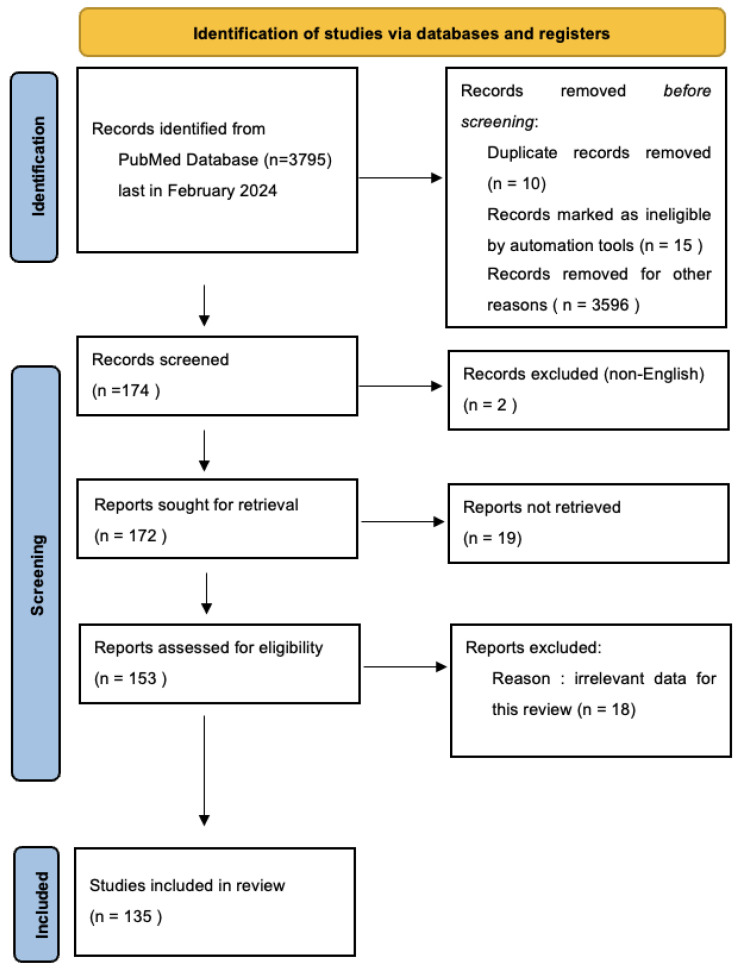
PRISMA flow diagram, 2020. Adapted from Page et al. [32].

**Table 1 ijms-25-02589-t001:** Level (increased/decreased) of selected cytokines in peripheral blood (b), CSF (c) or in MS lesions/intracranial (l) of subjects with MS/EAE.

	Increased	Decreased
IL-2	Petrzalka et al. (c) [37]; Gallo et al. (b) [39]; Sharief et al. (b, c) [40,41].	Carbone et al. (b) [43].
IL-4	Tahani et al. (b) [7]; Hulshof et al. (l) [59].	Ponomarev et al. (c) [58].
IL-6	Barr et al. (b) [63]; Patanella et al. (b) [65].	
IL-10		Ireland et al. (b) [60]; Rott et al. (l) [78].
IL-13	Ghezzi et al. (c) [89]; Rossi et al. (c) [90].	
IL-17	Schofield et al. (b, c) [94]; Tzartos et al. (l) [97].	
IL-21	Gharibi et al. (b) [109]; Christensen et al. (b) [110].	
IL-22	Muls et al. (b) [115]; Rolla et al. (b, c) [118]; Perriard et al. (b, l) [119]; Xu et al. (b) [120].	
IL-33	Alsahebfosoul et al. (b) [132]; Christophi et al. (b, l) [133].	

## Data Availability

Publicly available datasets were analyzed in this study.

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
