# Peer review of "The Role of Selected Interleukins in the Development and Progression of Multiple Sclerosis—A Systematic Review"

_ijms, 2024, doi:10.3390/ijms25052589_

Round 1

Reviewer 1 Report

Comments and Suggestions for Authors

The paper entitled The role of selected interleukins in the development and progression of multiple sclerosis – a systematic review brings new data about interleukins contributing to multiple sclerosis pathogenesis.

Observations

Introduction

line 36 - please enumerate the clinical forms of multiple sclerosis and their characters

- line 43 - please indicate the syndromes by which  the diagnosis of multiple sclerosis is suspicious - not only the symptoms (ataxia and so on..)

- line 69 - please give more details about the aim of this review (for example why this study is useful for further therapies).

- line 124 - please reformulate the sentence, because it is difficult to understand it.

Please write a table with pathogenetic role of interleukins in MS, according with your study.

Please explain in Introduction more details about the pathogenesis of BBB disruption and the role of this phenomenon in demyelnation and neurodegenerative process. Please include here the role of oxidative stress and inflammation, because the interleukines you are describing in this review are increasing during these processes. Describe the all molecules participating in this process and afterwards you can point it the role of interleukines. 

Reviewer 2 Report

Comments and Suggestions for Authors

In the review entitled “The role of selected interleukins in the development and progression of multiple sclerosis – a systematic review” the author provides a clear description of the literature data concerning the role of several interleukins involved in the regulation of the immune system during the development and the progression of multiple sclerosis disease.

It is extremely interesting to observe how several cytokines may play a protective role against the onset of the disease and how other molecules may enhance the onset and progression of the disease. Equally interesting is the complex mechanism by which some cytokines play a dual role, protective or not, on the onset of multiple sclerosis and/or during its progression.

The authors provide important insights into the possible study of molecular mechanisms activated by different cytokines in various autoimmune diseases, opening new avenues toward possible therapeutic strategies that could be used in neurodegenerative disorders.

The references reported are appropriate and include recent literature data, in line with the present study.

The following are some points that should be revised to better illustrate the present work.

-          In several lines, the references are placed incorrectly in the text, this may make it difficult to understand what exactly the inserted citation is referring to (see lines 35, 37, 146, 148, 150, 241,245, 307 and 314).

-          line 159, "invitro" should be corrected to "in vitro".

-          Those cytokines for which a dual mechanism has been observed, with both protective and promoting action on multiple sclerosis, should be specified in the conclusion section.

-          The doi should be included for reference 15, in the reference section.

-          The authors should consider including in the present work the role of IL2 and IL6, cytokines whose measured levels in the serum and CSF of MS patients have been described in a wide range of articles published in recent years. 
